# Exceptional figure of merit achieved in boron-dispersed GeTe-based thermoelectric composites

Yilin Jiang [1], Bin Su[1], Jincheng Yu [1], Zhanran Han[1], Haihua Hu[1], Hua-Lu Zhuang [1], Hezhang Li[1,2], Jinfeng Dong[1,3], Jing-Wei Li[1], Chao Wang[2], Zhen-Hua Ge[4], Jing Feng[4], Fu-Hua Sun[5] & Jing-Feng Li [1,4,5] ✉

GeTe is a promising p-type material with increasingly enhanced thermoelectric properties reported in recent years, demonstrating its superiority for mid-temperature applications. In this work, the thermoelectric performance of GeTe is improved by a facile composite approach. We find that incorporating a small amount of boron particles into the Bi-doped GeTe leads to significant enhancement in power factor and simultaneous reduction in thermal conductivity, through which the synergistic modulation of electrical and thermal transport properties is realized. The thermal mismatch between the boron particles and the matrix induces high-density dislocations that effectively scatter the mid-frequency phonons, accounting for a minimum lattice thermal conductivity of 0.43 $Wm^{-1}K^{-1}$ at 613 K. Furthermore, the presence of boron/GeTe interfaces modifies the interfacial potential barriers, resulting in increased Seebeck coefficient and hence enhanced power factor (25.4 $\mu Wcm^{-1}K^{-2}$ at 300 K). Consequently, we obtain a maximum figure of merit $Z_{max}$ of $4.0 \times 10^{-3} K^{-1}$ at 613 K in the GeTe-based composites, which is the record-high value in GeTe-based thermoelectric materials and also superior to most of thermoelectric systems for mid-temperature applications. This work provides an effective way to further enhance the performance of GeTe-based thermoelectrics.

Over half of heat energy is lost during the energy conversion process; the recovery of waste heat will be beneficial to environmental protection and economic development[1,2]. Thermoelectric (TE) materials enable the direct conversion between electricity and heat based on the Seebeck effect and Peltier effect, providing an eco-friendly route to resolve the problems raised above[3,4]. The dimensionless figure of merit, $ZT$, defined as $ZT = S^2\sigma T/\kappa$, where $\sigma$, $S$, $T$, and $\kappa$ are the electrical conductivity, Seebeck coefficient, absolute temperature, and total thermal conductivity, respectively, is an

essential indicator to evaluate the TE performance and conversion efficiency[5,6]. The $\kappa$ can be mainly divided into electronic thermal conductivity ($\kappa_e$) and lattice thermal conductivity ($\kappa_L$). Previous efforts made to improve $ZT$ values were mainly focused on two aspects: (i) the enhancement of the power factor ($PF = S^2\sigma$) via band manipulation[7,8] and carrier energy filtering[9], etc. and (ii) the reduction of $\kappa_L$ by introducing multiscale nanostructures, including point defects[10,11], dislocations[12,13], planar defects[14,15] and nano inclusions[16,17].

[1]State Key Laboratory of New Ceramics and Fine Processing, School of Materials Science and Engineering, Tsinghua University, Beijing 100084, China. [2]Department of Precision Instrument, Tsinghua University, Beijing 100084, China. [3]School of Materials Science and Engineering, Nanyang Technological University, Singapore 639798, Singapore. [4]Southwest United Graduate School, Kunming 650092, China. [5]Institute for Advanced Materials, Hubei Normal University, Huangshi 435002, China. ✉e-mail: jingfeng@mail.tsinghua.edu.cn

GeTe is one of the most promising lead-free compounds working in the medium temperature range. Due to the intrinsically high carrier concentration ($n_H \approx 10^{21}$ cm$^{-3}$) and the weak interaction between acoustic and optical phonons, pure GeTe shows low $S$ and high $\kappa_L$[18]. In earlier studies, the manipulation strategies of carrier transport can be mainly classified into two categories. One is the reduction of $n_H$ through Bi[19,20], Sb[21,22], etc. substitution for Ge. The other is the band structure modification by the use of dopants such as Zn[23], Cd[24], Pb[25], Cr[26], In[27], Ga[28] etc. Specifically, the former can indeed improve $S$ by reducing $n_H$, but inevitably deteriorates $\sigma$ to some degrees. Although the latter helps to enhance $S$ by increasing carrier effective mass ($m^*$), the carrier mobility ($\mu$) suffers from degradation due to the trade-off between $m^*$ and $\mu$. Consequently, the deteriorated $\sigma$ and increased $S$ only lead to slight changes in $PF$. On the other hand, although the atomic mass and ionic radius of the dopants are different from the host atoms, giving rise to the mass/strain fluctuations, and hence lower $\kappa_L$, these scattering centers only aim to scatter high-frequency phonons, showing limited effects on the reduction in $\kappa_L$ at low temperatures. Therefore, the large reduction in $\kappa_L$ usually requires complex compositions with the total content of foreign atoms over 10%, which also strongly scatter carriers and deteriorate $\sigma$ and $PF$[29]. As a result, the traditional doping methods usually yield modest $ZT$ values. Therefore, a new strategy needs to be developed to decouple the $n_H$ and $S$, and introduce other phonon scattering centers, with the aim of improving $PF$ and decreasing $\kappa_L$ in the GeTe system.

According to the Mott equation[30]:

$$S = \frac{\pi^2}{3} \frac{k_B}{q} (k_B T) \left[ \frac{1}{n(E)} \frac{dn(E)}{dE} + \frac{1}{\mu(E)} \frac{d\mu(E)}{dE} \right] \quad (1)$$

where $k_B$ is the Boltzmann constant and $q$ is the carrier charge, the TE performance can be optimized by introducing proper scattering sources. Notably, the interfaces can be rendered as a type of planar defect with its length scale in the order of nanometers to micrometers[31]. The additional heterogeneous interfaces can be easily introduced into the matrix materials by addition of nanoparticles. These appropriate heterogeneous interfaces hardly change the band structure of the matrix and $m^*$, but augment the scattering factor and hence $S$. On the other hand, they can act as new scattering centers for phonons, resulting in obvious reduction in $\kappa_L$. Successful paradigms of enhancing TE performance were achieved in SiC/(Bi,Sb)$_2$Te$_3$[9], B$_4$C/Cu$_2$Se[32], Nb/Mg$_3$(Sb,Bi)$_2$[4] and Co/Ba$_{0.3}$In$_{0.3}$Co$_4$Sb$_{12}$[33] systems, which show huge potential in decoupling the adversely inter-dependent $n_H$, $S$ and $\kappa$ in bulk materials.

According to our previous work, the boron inclusions play a significant role in minimizing $\kappa_b + \kappa_L$ in (Bi,Sb)$_2$Te$_3$ system[16]. Herein, the boron particles are incorporated into GeTe matrix materials with the aim of enhancing both the carrier and phonon scattering, because of their potential in modulating the interfacial barriers as well as microstructures, as shown in Fig. 1a. The interfacial barrier blocks part of holes, increasing the scattering factor and $S$, while the big difference in

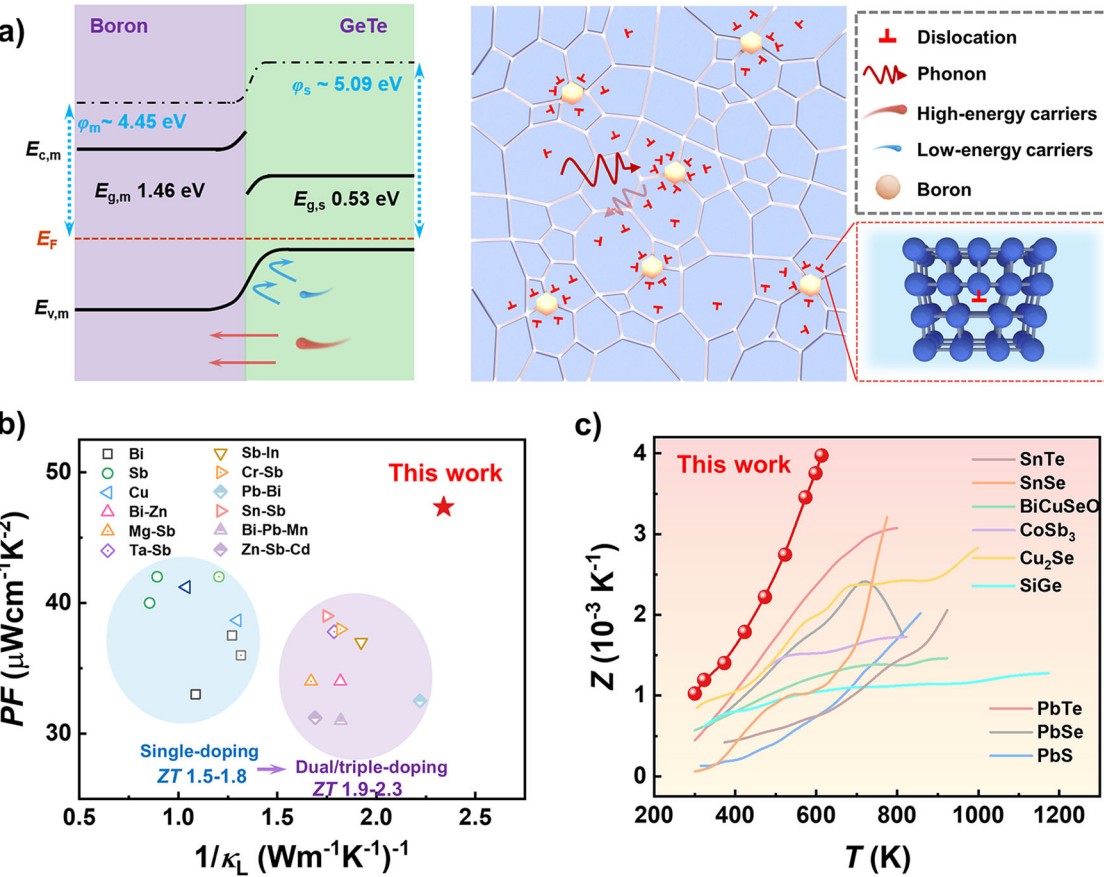

**Fig. 1 | Synergistic control of the carrier and phonon transports via interfacial strategy. a** The schematic image showing the positive effect on both carrier and phonons in boron/GeTe composites compared to the sample without boron[53]. **b** The comparison of the $PF$ and the corresponding value of $1/\kappa_L$ at the peak $ZT$ data point with different dopants: Bi doping[19,20], Sb doping[21,54], Cu doping[14,55], (Bi, Zn) co-doping[23], (Cd, Bi) co-doping[19], (Sb, Cr) co-doping[54], (Mg, Sb) co-doping[56], (Ta, Sb) co-doping[57], (Bi, Mn) co-doping[58], (Cr, Bi) co-doping[26], (Bi, Sc) co-doping[59], (Sn, Sb) co-doping[60], (Pb, Bi) co-doping[25], (Pb, Bi$_2$Te$_3$) alloying[61], (Zn, Sb, Cd) co-doping[62], (Bi, Pb, Mn) co-doping[63]. **c** The comparison of $Z$ values with different materials working in medium temperature: PbTe[64], PbSe[65], PbS[66], SnTe[67], SnSe[68], BiCuSeO[69], CoSb$_3$[70], Cu$_2$Se[71], SiGe[72].

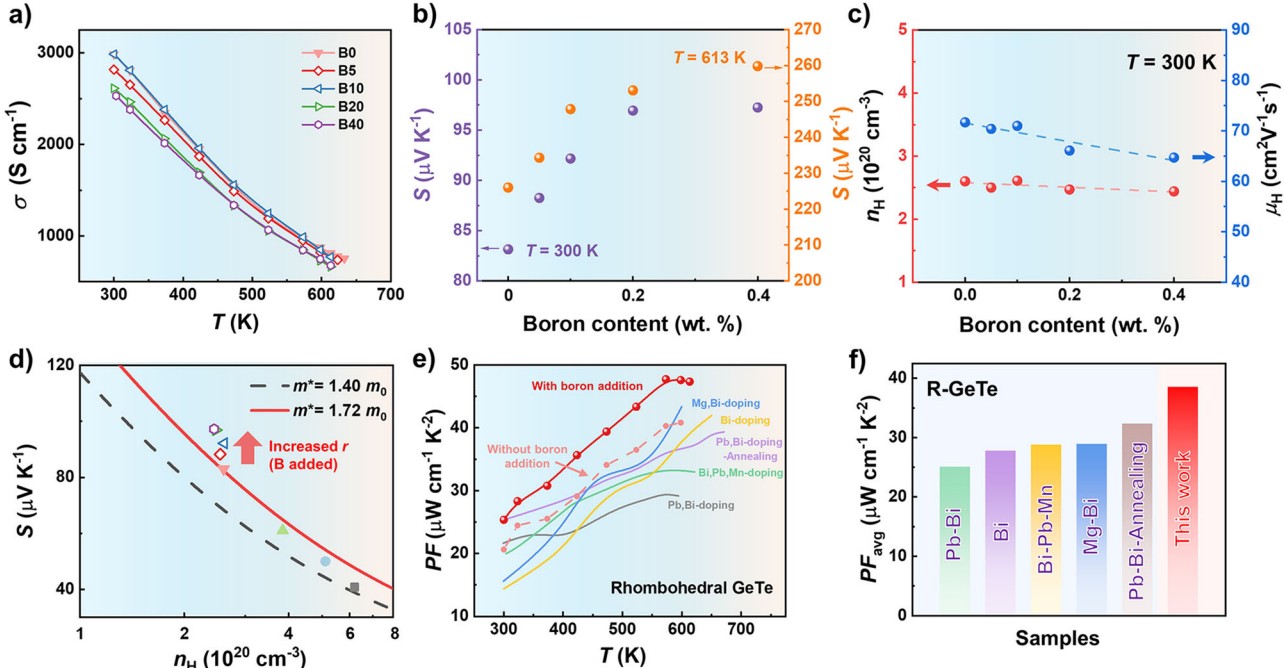

**Fig. 2 | The electrical transport properties. a** Temperature dependence of electrical conductivity for $Bi_{0.05}Ge_{0.94}Te$-$y$ wt. % B ($y = 0.00, 0.05, 0.10, 0.20, 0.40$) samples. **b** Seebeck coefficient of $Bi_{0.05}Ge_{0.94}Te$-$y$ wt. % B samples ($T = 300$ K and $T = 613$ K). **c** The effects of boron contents on carrier concentration and carrier mobility at 300 K. **d** The relationship between $S$ and Hall carrier concentration for $Bi_{0.05}Ge_{0.94}Te$-$y$ wt. % B samples at 300 K. The solid points represent for the Bi-doped GeTe samples. **e** Temperature dependence of power factor for B/BGT samples (B0/BGT and B10/BGT samples) and the comparison of power factor (for R-GeTe), and **f** average $PF$ with the values in literatures[20,29,42,48,62].

thermal expansion coefficient between boron and GeTe lead to the large strain fluctuations near the interfaces, inducing the formation of dislocations. As a result, the adversely dependent $n_H$ and $S$ are efficiently decoupled, leading to the enhanced $PF$ with maximum values of 25.4 $\mu Wcm^{-1}K^{-2}$ at 300 K and 47.7 $\mu Wcm^{-1}K^{-2}$ at 573 K, respectively. Furthermore, $\kappa_L$ is suppressed because the mid-frequency phonons are scattered by the strain-induced high-density dislocations. Due to the synergistic optimization of carrier and phonon transport, the boron-added samples obtain an extremely high $ZT$ value of 2.45 in R-GeTe compared to the samples prepared by the traditional doping methods (Fig. 1b). The maximum figure of merit ($Z_{max} = 4.0 \times 10^{-3}$ K$^{-1}$) of synthesized GeTe-based material is the record-high value in GeTe-based TE materials, and competitive among the TE materials for medium temperature applications, which is more intuitive to evaluate the transport properties without temperature factor (Fig. 1c, Supplementary Fig. 1). Further, a segmented single-leg TE device with a high conversion efficiency of 13.7% under a temperature gradient of 455.9 K was successfully fabricated based on the boron-dispersed GeTe composites. Our work sheds light on the interfacial engineering strategy to enhance the TE properties.

## Results

### Electrical transport

The temperature-dependent $\sigma$ and $S$ for Bi-doped GeTe samples are displayed in Supplementary Fig. 2. Bi is used to manipulate the carrier concentration here, reducing the $\sigma$ and improving $S$. Figure 2a, b and Supplementary Fig. 3a illustrate the temperature-dependent $\sigma$ and $S$ for $Bi_{0.05}Ge_{0.94}Te$-$y$ wt. % B samples ($y = 0.00, 0.05, 0.10, 0.20, 0.40$, namely B0/BGT, B5/BGT, B10/BGT, B20/BGT, B40/BGT). The $\sigma$ decreases with the increasing temperature. Meanwhile, the positive $S$ values show an opposite variation trend compared to $\sigma$, and indicate a typical p-type conducting mechanism (Fig. 2b and Supplementary Fig. 3). It is noteworthy that the $\sigma$ shows a downward trend with increasing boron content. Notably, the $S$ increases from 83.14 $\mu VK^{-1}$ to

97.3 $\mu VK^{-1}$ at 300 K as the boron content increases from 0 to 0.40 wt.%; the B0/BGT and B40/BGT samples reach maximum $S$ values of 226.5 $\mu VK^{-1}$ and 259.8 $\mu VK^{-1}$ at 613 K, respectively (Fig. 2b).

To gain a better understanding on the carrier transport behavior, the Hall measurement was conducted (Fig. 2c and Supplementary Table 1). With increasing boron content, the $n_H$ of B/BGT samples remains almost constant, whilst the carrier mobility shows a decreasing trend, resulting in slight reduction in $\sigma$. The relationship between $S$ and $n_H$ was further studied. Evidently, the solid line in Fig. 2d shows the Pisarenco curve for the B0/BGT sample ($x = 0.05$), supposing the scattering factor $r$ is $-0.5$. For a fixed $m^*$, the data points deviate upward largely at 300 K for the boron-added samples.

Benefiting from the significantly enhanced $S$, the $PF$ of the samples is boosted as shown in Fig. 2e and Supplementary Fig. 3. Overall, after boron addition, the $PF$s show distinguishable enhancement within the entire temperature range, especially in R-GeTe (before 613 K, investigated by DSC measurements, shown in Supplementary Fig. 4). Particularly, the $PF$ for the B10/BGT sample (25.4 $\mu Wcm^{-1}K^{-2}$) shows about 23.3% increase at 300 K, compared to that of B0/BGT sample (20.6 $\mu Wcm^{-1}K^{-2}$). The maximum $PF$ for the B10/BGT sample is augmented to 47.7 $\mu Wcm^{-1}K^{-2}$ in R-GeTe at 573 K. Furthermore, the comparison of temperature-dependent $PF$ with previously reported data is displayed in Fig. 2e. An outstanding $PF$ in the boron-added sample is achieved in R-GeTe. Our sample achieves a higher average $PF$ of 37.97 $\mu Wcm^{-1}K^{-2}$ over the temperature range in R-GeTe in comparison with other works, demonstrating greater output power potential for TE module fabrication (Fig. 2f). The comparison in PF and average PF values in the whole temperature range is shown in Supplementary Fig. 3. The transport properties after phase transition are shown in Supplementary Fig. 5.

### Thermal transport and ZT value

The temperature-dependent $\kappa$ and $\kappa_L$ of as-synthesized samples, along with the $\kappa_e$ determined by the Wiedemann–Frantz law is displayed in

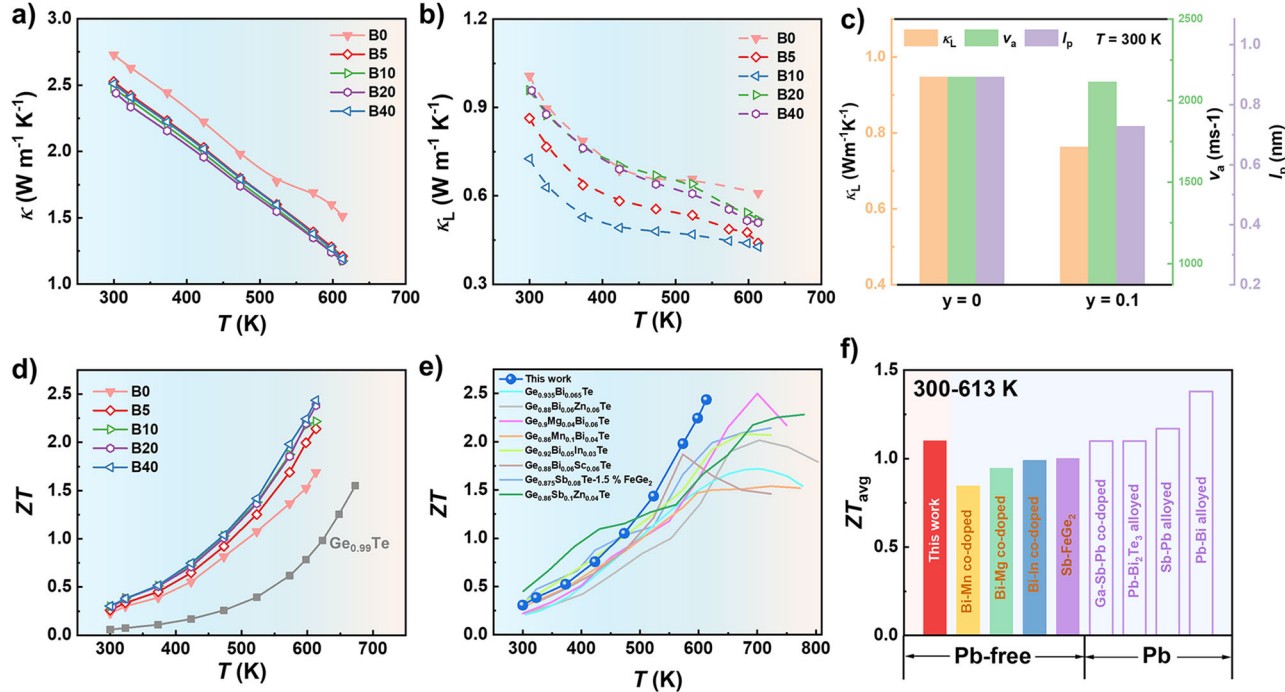

**Fig. 3 | Phonon transport properties and figure of merit ZT value.** Temperature dependence of **a** total thermal conductivity and **b** lattice thermal conductivity of $Bi_{0.05}Ge_{0.94}Te$-$y$ wt. % B samples. **c** The comparison of the B0/BGT and B10/BGT samples in the lattice thermal conductivity, sound velocity and mean free path of phonons. **d** ZT values of $Bi_{0.05}Ge_{0.94}Te$-$y$ wt. % B samples. Comparison of **e** the maximum ZT value and **f** the average ZT value of GeTe-based materials in this work with lead-free TE materials[20,23,48,58,59,73–75] and Pb-doped TE materials[25,42,76,77].

Fig. 3a, b and Supplementary Fig. 6, respectively. The $\kappa_L$ is obtained by subtracting $\kappa_e$ from $\kappa$. The $\kappa$ of the boron-added samples decreases within the entire temperature range compared to the B0/BGT sample (Fig. 3a); the $\kappa_e$ of the boron-added samples decreases, attributed to the slightly reduced $\sigma$. Fig. 3b demonstrates the variation of $\kappa_L$ for samples with different boron contents; it is clear that $\kappa_L$ shows a decrease followed by a rise as the boron content increases. Fig. 3c shows that the main reason for the decreased $\kappa_L$ should be the reduced mean free path of phonons ($l_p$), demonstrating the enhanced phonon scattering caused by boron addition. As a result, a high ZT value of 2.45 is achieved in the rhombohedral B10/BGT sample at 613 K (Fig. 3d). Notably, our work also demonstrates good repeatability (ZT = 2.44 and $Z = 4.0 \times 10^{-3}$ K$^{-1}$ in Supplementary Fig. 7), showing much higher ZT values than most of other GeTe-based TE materials (Fig. 3e). The specific heat capability $C_p$ values for the sample is measured as shown in Supplementary Fig. 8. In addition, boron-added samples attain a high average ZT of 1.1 in the temperature range from 300–613 K, which ranks at a high level among lead-free GeTe-based materials and is comparable to most of lead-doped GeTe systems (Fig. 3f).

**Phase and microstructure characterization**

In order to clarify the regulatory mechanisms determining the electrical and phonon transport properties, the phase and microstructures of the as-prepared samples were investigated. The powder X-ray diffraction (PXRD) was used to investigate the crystal structure of B/BGT samples, as shown in Supplementary Fig. 9. The prominent peaks for all the samples correspond well to the rhombohedral GeTe in $R3m$ space group, and the peaks of boron is undetectable here because of the low additive content. The lattice parameters were calculated by XRD Rietveld refinement, shown in Supplementary Fig. 10. The lattice parameter $a$ and interaxial angle $\alpha$ are insensitive to boron addition (Supplementary Table 2). The in-situ high-temperature XRD is shown in Supplementary Fig. 11. To investigate the elemental distribution in the matrix, the electron probe micro-analysis (EPMA) and back-

scattered electron imaging (BEI) were carried out (Supplementary Figs. 12–13). It is found that the trace of boron inclusions can be detected in the boron-added samples, along with the generally observed Ge precipitates. Furthermore, the field-emission scanning electron microscopy (FESEM) images of the fracture morphology are shown in Supplementary Fig. 14. It is also evident that the grain size decreases as the boron content increases, as a result of the Zener pinning effect[34]. This finding is consistent with the electron backscatter diffraction (EBSD) analysis (Supplementary Fig. 15).

The detailed structural information about the boron inclusions and matrix materials were further examined by scanning transmission electron microscopy (STEM) as shown in Supplementary Fig. 16. Energy-dispersive X-ray spectroscopy (EDS) mapping confirms the presence of boron inclusion and the uniform distribution of Bi, Ge and Te in the matrix. Impressively, the boron inclusions in the size of several tens to hundreds of nanometers are present accompanied by high-density dislocations (Fig. 4b, c and Supplementary Figs. 17–19) compared to the sample without boron addition (Fig. 4a). As shown in Supplementary Fig. 18, the selected area electron diffraction (SAED) pattern of boron inclusions viewed in the [010] zone axis shows that the corresponding space group is indexed as $R$-$3m$ (space group no. 166)[35].

The interfacial contact between boron inclusions and GeTe matrix and the dislocations was further investigated by high-resolution TEM (HRTEM). As shown in Fig. 4d, the incoherent interface between boron and matrix is manifested. The HRTEM image (Supplementary Fig. 18b), and the fast Fourier transformation (FFT) images (Supplementary Fig. 18d–e) also indicate that there is no orientation relationship between the matrix and inclusions. Supplementary Fig. 20 shows another typical incoherent interface, the FFT image of which reflects the diffraction spots assigned to GeTe matrix in the [001] zone axis (marked in yellow) and the boron inclusions in [010] zone axis (marked in red). Furthermore, the inverse fast Fourier transformation (IFFT) was used to identify the dislocation structure and distorted lattice in

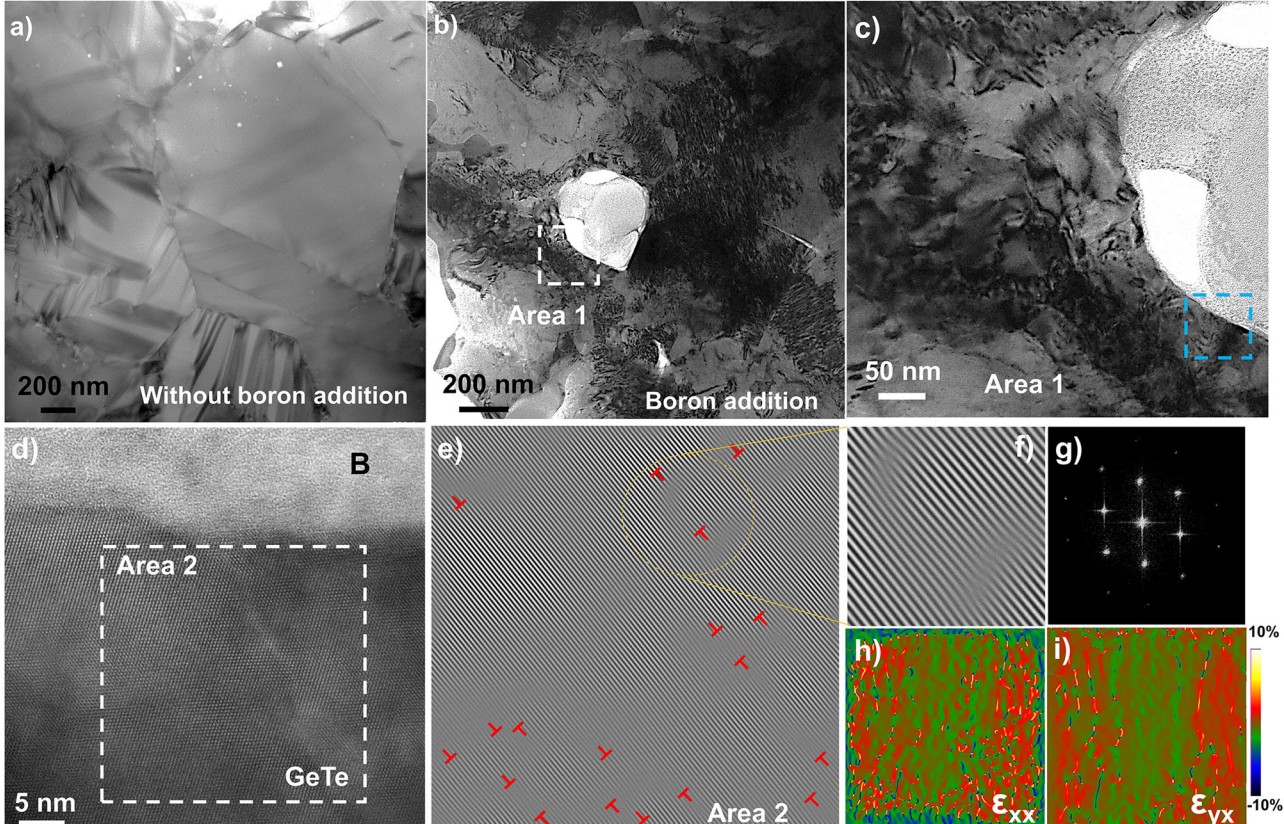

**Fig. 4 | Microstructure evolution led by boron addition.** The low magnification transmission electron microscopy (TEM) for **a** the sample without boron addition and **b** the sample with boron addition. **c** The enlarged area (area 1) in **b**. **d** The HRTEM image showing the interface between the boron inclusion and GeTe matrix. The corresponding **e, f** IFFT images **g** FFT image showing the area 2 in **d** indicating the dislocations in GeTe matrix. Strain mapping along **h** xx direction and **i** xy direction confirmed by geometric phase analysis (GPA).

Fig. 4e–f and Supplementary Figs. 18–19, indicating high-density dislocations at the interface. Supplementary Fig. 21 even shows deformation twinning structures at the interfaces, in addition to dislocations. It is also noted that the high-density dislocations were mainly distributed at the heterogeneous interfaces, the density of which shows an obvious reduction away from the boron inclusions (Supplementary Fig. 22).

Here, the presence of high-density dislocations around the inclusions can be ascribed to the difference in thermal expansion coefficients (TEC) between boron (around $1–5 \times 10^{-6}$ K$^{-1}$ (in *a* axis) and $4–8 \times 10^{-6}$ K$^{-1}$ (in *c* axis) in 300–873 K)[36] and GeTe ($5.60 \times 10^{-5}$ K$^{-1}$ in 296–648 K and $5.78 \times 10^{-5}$ K$^{-1}$ in 648–948 K (volume TEC))[37]. By the Eshelby's inclusion model[38,39], the misfit strain $\varepsilon$ can be calculated by the following formula:

$$\varepsilon = (\alpha_M - \alpha_I) \cdot \Delta T \tag{2}$$

where $\alpha_M$ and $\alpha_I$ is the TEC of matrix and inclusions, respectively, and $\Delta T$ is the temperature drop. Supplementary Fig. 23 shows the difference of TEC, and the strain here is calculated to be 2.5 %. The strain generated during the sintering process drive the evolution from vacancies to dislocations in $Bi_{0.05}Ge_{0.94}Te$ (with 6 at. % Ge deficiency in theory)[40,41]. Figure 4h–i show the strain maps along different directions, revealing large strain fluctuation near the interfaces. Essentially, these dislocations can also act as effective sources to scatter mid-frequency phonons. By determining the crystal structure, particle distribution and interface condition of boron inclusions, we can gain deeper insights into the influence of crystalline boron on carrier and phonon transports.

## Mechanism analysis

To better understand the change in electrical transport properties, the DFT calculations were carried out to examine the electronic band structure via Bi doping (Supplementary Fig. 24). Both the s orbital energy of the dopants and the interaxial angle are key to inducing the band convergency[42]. According to the previous results[23], the s orbital energy of Bi does not obviously contribute to the band convergency. Considering the changes in crystal structure induced by Bi (Supplementary Table 2), the valance band convergency is promoted, supported by our DFT calculation results. After Bi doping, the energy separation between the valence band maxima at the L and $\Sigma$ points decreases from 174 meV to 74 meV. As a result, Bi doping improves the $m^*$ and $S$. As for boron addition, there are negligible effects on lattice parameters (especially the interaxial angle, Supplementary Table 2), and boron atoms hardly enters the matrix lattice. Therefore, the interface should be responsible for the phenomenon in Fig. 2d. Figure 1a shows the interfacial band diagram of the boron and GeTe, showing the work functions of two materials (GeTe: -5.09 eV[43], boron: -4.45 eV[44]). The interface contact between the matrix and the boron inclusion is exactly a p-p homojunction. As the work function of GeTe is higher than that of boron, the electrons tend to transfer from boron to GeTe, resulting in an internal electric field pointing from boron to GeTe. Consequently, a depletion layer is revealed at the interface[45]. This depletion layer can perform as an interfacial potential barrier to block part of holes and modulate the carrier scattering factors.

$$S = [8\pi^2 k_B^2 / (3eh^2)] m^* T [\pi/(3n)]^{2/3} (r + 3/2) \tag{3}$$

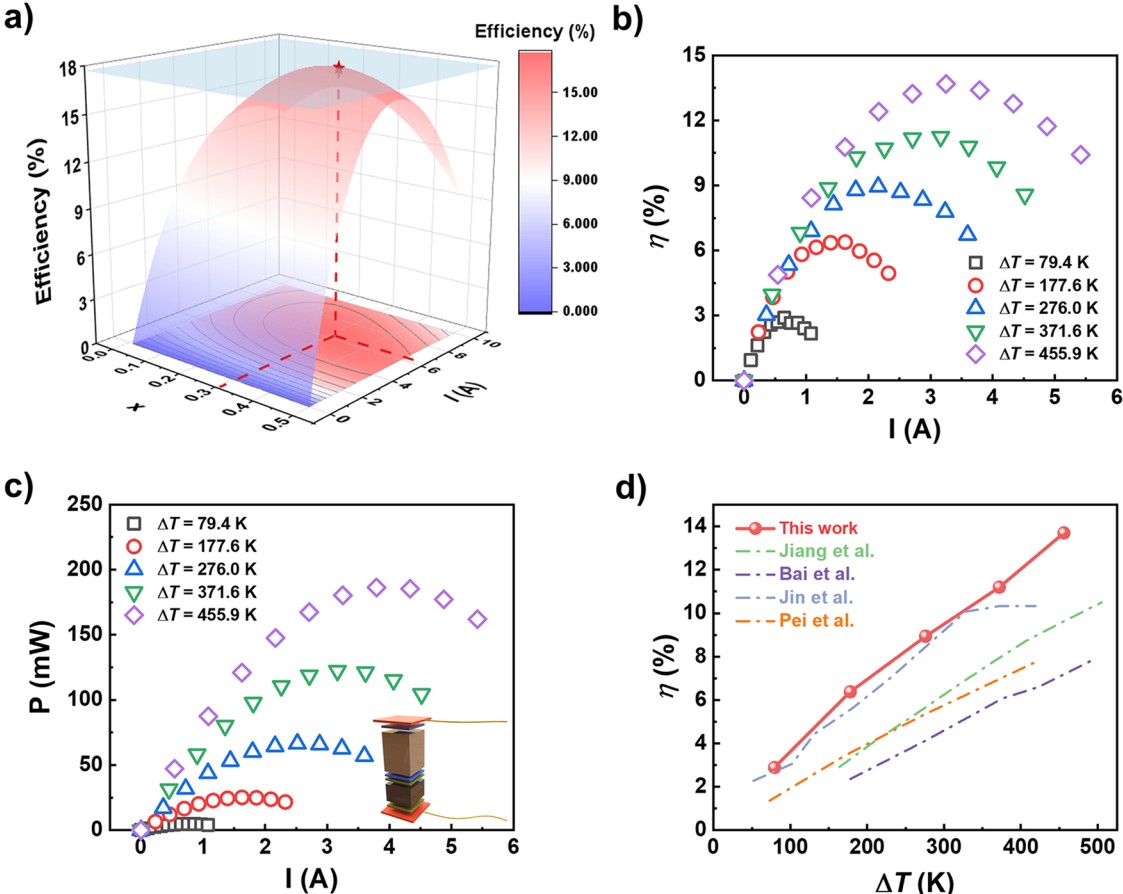

**Fig. 5 | The stimulation and measurement results of the single-leg thermo-electric device. a** Contour map of efficiency ($\eta$) of GeTe/(Bi,Sb)$_2$Te$_3$ segmented TE leg when $T_h$ = 723 K and $T_c$ = 300 K. **b, c** The tested conversion efficiency and output power, respectively. The inset image in **c** showing the schematic diagram of the segmented TE leg. **d** The comparison of the conversion efficiency with the results in literatures[17,43,51,52].

is used to further figure out the carrier scattering factor, where $e$ is the electron charge, $h$ is the Planck constant and $n$ is the carrier concentration[46]. The carrier scattering in the Bi-doped sample (the B0/ BGT sample) is usually dominated by the acoustic phonons. Thus, $r$ is fixed to be −0.5, and the $m^*$ can be determined to be 1.72$m_0$ based on the SPB model. As for the boron-added samples, the $r$ can be estimated given a fixed $m^*$, and the $r$ can be figured out, shown in Supplementary Table 3.

But the interfacial potential barrier also leads to a slight decrease in the mobility of B/BGT samples, especially in the sample with higher boron content. From the aspect of Mott equation, the increased $S$ should be assigned to changes in carrier energy dependent mobility. From the microscopic point of view, $S$ can be defined as the heat or more simply the entropy per carrier[47]. These inclusions generate abundant interfaces, which helps to enhance the carrier scattering, leading to larger entropy and hence increased $S$. It is found that the similar phenomenon is also observed in the boron-added samples sintered at 723 K (Supplementary Fig. 25).

Furthermore, the reduction in $\kappa_L$ (Fig. 3b) can be attributed to the enhanced phonon scattering, deducted from the presence of strain-induced dislocations and boron inclusions (Fig. 4). The rise in $\kappa_L$ might be attributed to the nonuniform distribution of excessive boron particles and their thermally conductive nature. In order to figure out the main reason for reduced $\kappa_L$, the Bi$_{0.05}$Ge$_{0.96}$Te and Bi$_{0.05}$Ge$_{0.96}$Te-0.1 wt. % B samples (less Ge deficiency samples) were fabricated. Supplementary Fig. 26 shows the cation-excessive sample have slight variations in $\kappa_L$ value after boron addition, while the boron-added samples with more Ge deficiencies show remarkable

reduction in the $\kappa_L$ value, which can be assigned to the high-density dislocations at the interfaces. The regions enriched with high-density dislocations show higher Ge deficiencies (Supplementary Figs. 26–27), manifesting the role of Ge content in inducing dis-locations. Furthermore, the boron-added sample with the same composition sintered at 723 K shows negligible variations in $\kappa_L$ as shown in Supplementary Fig. 28. It is found that there are few dis-locations in the matrix or near the boron inclusion in Supplementary Fig. 29. Through the comparison of the samples with the same boron content sintered at different temperatures (Fig. 3 and Supplementary Figs. 28–29), it can be deducted that the primary contribution to the reduction in $\kappa_L$ is the formation of dislocations. Because the two samples have the same amount of boron inclusions, the formation of dislocations should be mainly attributed to the sintering tempera-ture. On one hand, the higher sintering temperatures may promote the evolution of vacancies. On the other hand, the higher sintering temperatures may induce higher strains due to the higher tempera-ture drop. Consequently, a minimum $\kappa_L$ of 0.43 Wm$^{-1}$K$^{-1}$ is achieved in the B10/BGT sample, approaching the theoretical minimum $\kappa_L$ of GeTe following the Clarke model[48,49].

To further clarify the contributions from inclusions and high-density dislocations, the $\kappa_L$ was fitted by the Debye−Callaway model (shown in "Supplementary Materials" section). Here, $\kappa_L$ can be calcu-lated from the following equation[50]:

$$\kappa_L = \frac{k_B}{2\pi^2 \nu_s}\left(\frac{k_B T}{\hbar}\right)^3 \int_0^{\theta_D/T} \tau_{tot}\frac{z^4 e^z}{(e^z-1)^2}dz \quad (4)$$

The integrand item, in conjunction with the coefficient of Eq. 4, is the spectral lattice thermal conductivity ($\kappa_s$), namely:

$$\kappa_s = \frac{k_B}{2\pi^2 \nu_s} \left(\frac{k_B T}{\hbar}\right)^3 \tau_{tot} \frac{z^4 e^z}{(e^z - 1)^2} \tag{5}$$

where $k_B$ is the Boltzmann constant, $\nu_s$ is the average sound speed, $\hbar$ is the reduced Plank constant, $\theta_D$ is the Debye temperature, $z = \hbar\omega/k_B T$ ($\omega$ represents the phonon frequency) is the reduced phonon frequency and $\tau_{tot}$ is the total relaxation time.

The phonon scattering mechanisms of the Umklapp process (U), grain boundaries (B), point defects (PD), dislocations (D), and precipitates (P) were taken into account based on the microstructure characterization (Supplementary Fig. 30). The pinning effect here leads to a decrease in grain size, which restricts the propagation of low-frequency phonons, whilst the point defects are the predominant factor for high-frequency phonons. Furthermore, $\kappa_L$ is suppressed because the mid-frequency phonons are scattered by strain-induced high-density dislocations. According to the calculation results, the dislocations play a major role in scattering the mid-frequency phonons compared to the inclusions, which is consistent with our observations.

## Fabrication and evaluation of TE device

A segmented single-leg thermoelectric device was designed by integrating $(Bi,Sb)_2Te_3$ with GeTe in view of their excellent TE performance within different temperature range (Fig. 3 and Supplementary Fig. 31). The output power density and conversion efficiency of the segmented single-leg device were simulated as a function of the current and the height ratio via the finite element method[43]. The total height of the single-leg was set as 9 mm, where the height ratio of $(Bi,Sb)_2Te_3$ to GeTe was defined as $x/(1-x)$. According to our calculation results (Fig. 5a), the maximum TE conversion efficiency is 17.7% when $x = 0.30$ and $I = 6.4$ A. To achieve a higher TE conversion efficiency, the value of $x = 0.30$ was selected for the device fabrication.

Finally, a segmented single-leg TE device with a height ratio of 3:7 was fabricated, as illustrated in Supplementary Fig. 32. Ti and Ni were employed as the metallized layers linking to GeTe and $(Bi,Sb)_2Te_3$, respectively. The copper electrode is then connected with the device via soldering. As shown in Fig. 5b, the highest conversion efficiency ($\eta_{max}$) of 13.7% is yielded ($\Delta T = 455.9$ K). When $\Delta T = 455.9$ K and $I = 3.79$ A, the maximum output power exceeds 0.18 W (Fig. 5c), and the corresponding V–I relationship and heat flow are shown in Supplementary Fig. 33. Figure 5d indicates the comparison of conversion efficiency in this work with previous studies[17,43,51,52]. However, the measured values still deviate from the calculated results, indicating that further optimization of the metallization layer and fabrication process is needed.

In summary, this work demonstrates the synergistic optimization of electrical and phonon transport properties via interfacial engineering strategy in the boron/GeTe composites, improving the TE performance of the R-GeTe. Boron/GeTe heterogeneous interfaces prove effective in scattering carriers, increasing the carrier entropy, and hence enhancing $r$ and $S$. As the $\sigma$ does not suffer from degradation, $PF$ of the B/BGT samples are significantly improved. In particular, due to the improvement of $PF$ in R-GeTe, the B10/BGT sample exhibit a high average $PF$, which is critical for the improvement in output power density of the TE module. Additionally, the incoherent interfaces between the matrix and the inclusions enhance the phonon scattering. The great difference in TEC between boron inclusions and GeTe leads to the large strain around the interfaces, inducing the evolution of dislocations, which play a major role in scattering mid-frequency phonons. The $\kappa_L$ is reduced to 0.43 Wm$^{-1}$K$^{-1}$ at 613 K in the B10/BGT sample. Consequently, a maximum $ZT$ value of 2.45 is achieved. An average $ZT$ value of 1.1 is also obtained in the B10/BGT sample within the temperature range of 300–613 K. Moreover, the as-prepared GeTe-$(Bi,Sb)_2Te_3$ segmented single-leg TE device shows a high energy conversion efficiency of 13.7%.

## Methods

### Sample fabrication

Raw materials, germanium (granules, 2–5 mm, 99.999%, ZhongNuo Advanced Material (Beijing) Technology Co., Ltd), tellurium (powder, 99.999%, ZhongNuo Advanced Material (Beijing) Technology Co., Ltd), and bismuth (powder, 99.99%, Aladdin) were weighed in the glove box and loaded into tungsten carbide jars according to the stoichiometric ratios of $Bi_x Ge_{0.99-x} Te$ ($x = 0, 0.01, 0.03, 0.05$). First, the mixture (Bi, Ge and Te) was reacted via mechanical alloying (MA) in a planetary ball mill at 450 rpm for 10 h, with argon ( > 99.5%) as the protective gas. Next, the powders ($Bi_{0.05} Ge_{0.94} Te$) were mixed with amorphous boron (powder, 99%, Aladdin) via MA in a planetary ball mill at 300 rpm for 2 h to fabricate a series of boron-added samples ($Bi_{0.05} Ge_{0.94} Te$-$y$ wt. % B samples, in which $y = 0.00, 0.05, 0.10, 0.20, 0.40$). The total mass of powders for one jar is 10 g, in which the addition amount of boron powder is 0 g, 0.005 g, 0.01 g, 0.02 g and 0.04 g for $Bi_{0.05} Ge_{0.94} Te$-$y$ wt. % B samples where $y = 0.00, 0.05, 0.10, 0.20, 0.40$, respectively. Then, the obtained powders (about 7 g) were densified by spark plasma sintering (SPS 211Lx, Fuji Electronic, Japan) at 873 K for 5 min under a pressure of 60 MPa.

### Characterization

We used X-ray diffraction (D8 ADVANCE, Bruker, Germany, Cu Kα, $\lambda = 1.5418$ Å) to identify the phase purity of samples. The field-emission scanning electron microscopy (Zeiss Merlin, Germany), and transmission electron microscopy (2100 F, JEOL, Japan) were used to investigate the grain morphology and microstructure. We used electronic probe microscopic analysis (JXA-8230, JEOL, Japan) to study the elemental distribution of samples.

### Transport properties measurement

The obtained boron-added GeTe samples were cut into bars with dimensions of ~$2.5 \times 2.5 \times 9$ mm$^3$, used for the measurements of the Seebeck coefficient and electrical conductivity via the measuring system (Ulvac Riko ZEM-3, Japan) under a helium atmosphere from room temperature to 723 K. The obtained samples were cut into disks with dimensions of $\varphi$ ~ 6 mm and thickness of ~1 mm. The disks were coated with a thin layer of graphite for thermal diffusion coefficient (D) measurements using the laser flash method (LFA457, Netzsch, Germany). The thermal conductivity was calculated according to $\kappa = D C_p \rho$, where $C_p$ is the specific heat, and the density ($\rho$) was measured by Archimedes' method. The $C_p$ value was deduced via the Dulong-Petit limit, which was used for calculating the thermal conductivity. In Supplementary Fig. 8, the $C_p$ value was measured by differential scanning calorimetry (STA 449 F3 Jupiter, Netzsch, Germany) with a heating rate of 5 K/min. We used the Wiedemann-Franz law $\kappa_e = \sigma LT$ to calculate the electrical thermal conductivity, where the Lorenz factor (L) was estimated according to the formula $L = 1.5 + \exp(-|S|/116)$. The samples we used for the measurement of the electrical and thermal transport properties were perpendicular to the axial SPS pressure. The obtained samples were cut into pieces with dimensions of $10 \times 10 \times 0.5$ mm$^3$, used for Hall coefficient ($R_H$) measurements (ResiTest 8340DC, Japan). We calculated the Hall carrier concentration ($n_H$) and mobility ($\mu_H$) according to the formula $n_H = 1/(eR_H)$ and $\mu_H = \sigma R_H$, respectively. We used the ultrasonic pulse-echo technique (5072PR, Olympus, Japan) to measure the sound velocity (v).

### The fabrication and characterization of thermoelectric device

The thermoelectric single-leg module was assembled in a glove box and sintered by SPS. Ti and Ni is employed as the metallized layer at the side of GeTe and $(Bi,Sb)_2Te_3$, respectively. The copper electrode is connected via soldering. We calculated the energy conversion efficiency ($\eta$) of the segmented single-leg device according to the equation $\eta = P/(P + Q) \times 100\%$, where the output power (P) and heat flow per unit

time ($Q$) were measured by the commercial Thermoelectric Conversion Efficiency Evaluation System for Small Modules (Mini-PEM, Advance Riko, Japan). Due to the small size of the device and high testing temperature, the $Q$ was revised according to the analysis system provided by the company. We used the COMSOL Multiphysics software to simulate and optimize the theoretical conversion efficiency of the single-leg device.

## Data availability
The data that support the findings of this study are available from the corresponding author on request.

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

## Acknowledgements

J.F.L., Y.L.J., B.S., J.C.Y., Z.R.H., J.F.D., H.L.Z., H.Z.L., H.H.H., J.W.L., J.F., F.H.S., Z.H.G. and C.W. acknowledge the National Key R&D Program of China (2023YFB3809400) and the Basic Science Center Project of National Natural Science Foundation of China (grant no. 52388201).

## Author contributions

J.F.L. supervised this work. Y.L.J. synthesized the samples and carried out the transport property measurements. Y.L.J., J.C.Y. and H.L.Z. analyzed the XRD results and carried out Rietveld refinement. C.W. carried out the in-situ high-temperature XRD analysis. Y.L.J., B.S., H.Z.L. and J.F.D. performed the TEM observations. Z.R.H. and Z.H.G. conducted the DFT calculations. J.F. and Y.L.J. performed the DSC tests. J.F.L., B.S., Y.L.J., H.H.H., J.W.L. and Z.R.H. discussed the mechanisms. Y.L.J. wrote the manuscript and J.F.L., J.C.Y., J.F.D., H.L.Z. and F.H.S. revised the manuscript.

## Competing interests

The authors declare no competing interests.
