## [Peer Review File · Nature Communications]

Exceptional figure of merit achieved in boron-dispersed GeTe-based thermoelectric compositesREVIEWER COMMENTS

Reviewer #1 (Remarks to the Author):

This work reports a very impressive figure of merit Z value of $4.0 \times 10^{-3} \text{ K}^{-1}$ in GeTe system. The authors systematically investigate the effects of boron particles on electrical and phonon transport properties. The heterogeneous interfaces successfully block part of carriers, improving the Seebeck coefficient and PF. Moreover, the detailed TEM and STEM images provide clear insights into the interfaces and the high-density dislocations, which helps to understand the reduced lattice thermal conductivity easily. Overall, this manuscript is well organized and the data are sufficient to support the final conclusions. This nice work opens an avenue to achieve the synergistic optimization of both the electrical and thermal transports. Therefore, I recommend this manuscript for publication in Nature Communications after addressing my minor concerns.

1. The authors emphasized the electrical transport properties of the rhombohedral GeTe in fig.2. It is suggested that DSC data should be collected to verify the accurate phase transition temperature.
2. In fig.3, the thermal conductivity decreases as the boron content increases. However, the lattice thermal conductivity decreases first, followed by a rise. Could you please explain why the lattice thermal conductivity exhibits such a variation trend?
3. Please provide the corresponding voltage-current (V-I) plot to support the output power and efficiency data of the device.
4. It is evident that there is a difference in the energy conversion efficiency between the simulation data and experimental data of the device. Could you please give some comments?

Reviewer #2 (Remarks to the Author):

The manuscript reports on the thermoelectric properties of the (Ge,Bi)Te + B samples. While there are a lot of good data, the data analysis and representation are of sufficient quality for the Nature Communication.

1. The title itself and abstract are misleading. First of all, the samples are not nanocomposites. The prepared samples are polycrystalline, with the submicrometric inclusions of B and Ge impurities ($>200 \text{ nm}$). Secondly, GeTe is doped with Bi, which improves the properties significantly. This is not stated in the title nor mentioned in the abstract; such omission creates an illusion that B does all the work.
2. Analysis of the scattering parameter, r , in Eq. 3 on page 11 is questionable. Both the r and effective mass, m^* , will influence the Seebeck coefficient, and thus one cannot establish r without knowing m^* . The authors refer to Ref. 73, where this issue of uncertainty of r and m^* was stated. The authors appeared to ignore this and decided to derive r from S , without really stating what value of m^* was used and why.
3. The choice of the additional sintering temperature of 723K is not clear. Yes, this temperature will not promote Ge dislocation as the temperature is low and external pressure is large.

Despite this, the data in Fig. S 22 and 23 for the BGT-723 and BGT/B40-723 are very useful especially in comparison to the data in Fig 3b and 4 for BGT and BGT/B40. From this comparison, one can deduce that the primary contribution to the reduction in the lattice thermal conductivity is formation Ge dislocations, and not the B inclusions.

Both samples have the same amount of B inclusions and the only difference between them

is the annealing temperature and the resulting Ge vacancy dislocations. Thermal mismatch between the boron particles and the GeTe matrix should be relatively similar for the two sintering temperatures.

4. The entire narrative of the low thermal conductivity due to the B inclusions is not valid. Other researchers reported similar thermal conductivities in the samples that do not have B inclusions. For example, <https://doi.org/10.1002/adma.202008773>, <https://doi.org/10.1021/jacs.8b12624>.

5. There is no convincing argument proving that the increase in thermopower stems from the “the interfacial potential barriers” and not from the increase in the carrier effective mass related to the changes in the electronic structure. The Pisarenko plot in Figure 2d points to the increase in the effective mass. In the same figure, the authors indicate that the scattering parameter r increases, but as discussed above this is questionable.

6. Figure 2e. Please, show data for the $\text{Ge}_{0.9}\text{Mg}_{0.04}\text{Bi}_{0.06}\text{Te}$ sample till 700K. <https://doi.org/10.1002/adma.202008773>

7. Figure 3e. Please, include the data for $\text{Ge}_{0.86}\text{Sb}_{0.1}\text{Zn}_{0.04}\text{Te}$. <https://doi.org/10.1021/jacs.8b12624>

8. $\text{Bi}_{0.05}\text{Ge}_{0.94}\text{Te}$ and $\text{Bi}_{0.05}\text{Ge}_{0.94}\text{Te}+\text{B}$ are also cation excessive. The charge balanced composition is $\text{Bi}_{0.05}\text{Ge}_{0.925}\text{Te}$.

9. The authors should report on the amount of each sample made. This is important in the light of the B amount used.

Reviewer #3 (Remarks to the Author):

In the manuscript by Jiang et al., a high thermoelectric figure of merit (zT) for a GeTe-based composite is reported, along with impressive one-leg conversion efficiency, attributed to an extremely modest addition of boron. Despite the noteworthy results, the employed strategy of nanocompositing and the claim of Bi-GeTe achieving a peak zT exceeding 2 are not novel. Consequently, the work may not be suited for a high-profile publication like Nature Communications but could find a home in a more specialized materials science journal. Before considering resubmission, the authors should resolve some inconsistencies:

1. The manuscript emphasizes the role of Bi as a phonon scattering center that alters band mass. If so, detailed theoretical band structure calculations, especially changes in effective mass after Bi incorporation, should be presented. The effective mass derived from the Pisarenko plot in Figure 2d appears to be merely an empirical fit.

2. While the carrier concentration remains constant at 300 K (Figure 2c), a notable decrease in the Seebeck coefficient for B0 and B5 at the same temperature is observed. Clarification on this discrepancy is required.

3. The zT curves for B-GeTe (Figure 3d) deviate from typical behavior in GeTe systems. The peak values coincide with phase transition temperatures, raising concerns about structural stability, as the highest performance is precariously near the phase transition. Moreover, there is a precipitous performance drop after the peaks are reached.

4. The claim of achieving a minimum lattice thermal conductivity (KL) of $0.37 \text{ Wm}^{-1}\text{K}^{-1}$ in the B10/BGT sample, nearing the theoretical minimum KL for GeTe, necessitates verification via measured heat capacity data, particularly because of the boron composite's nature.

5. When assessing conversion efficiency, the potential for thermal radiation losses during heat flow measurement with mini-pem must be considered and discussed, as this is a common measurement challenge.

6. Supplementary Figure S21e & f present Ge concentrations of 28 & 36 at% Ge, which seem implausible. It is recommended to corroborate these findings with low magnification SEM images in backscattered electron (BEI) mode, supplemented by WDS analysis to

authenticate elemental distribution.

7. Inclusion of high-temperature (in-situ) powder X-ray diffraction (PXRD) data for the best-performing samples is advisable, given that the optimal performance region is significantly above room temperature.

Dear Editor and reviewers,

Thank you very much for your time and efforts in handling our manuscript entitled: **“Exceptional figure of merit achieved in boron-dispersed GeTe-based thermoelectric composites”** (Manuscript ID: NCOMMS-24-11820). We greatly appreciate the reviewers’ valuable comments and suggestions. We hope the revised manuscript will address the concerns of the reviewers and meet the requirements of your esteemed journal *Nature Communications*. Our point-to-point responses are listed as below:

Answers to reviewers:

Reviewer #1 (Remarks to the Author):

This work reports a very impressive figure of merit Z value of $4.0 \times 10^{-3} \text{ K}^{-1}$ in GeTe system. The authors systematically investigate the effects of boron particles on electrical and phonon transport properties. The heterogeneous interfaces successfully block part of carriers, improving the Seebeck coefficient and PF. Moreover, the detailed TEM and STEM images provide clear insights into the interfaces and the high-density dislocations, which helps to understand the reduced lattice thermal conductivity easily. Overall, this manuscript is well organized and the data are sufficient to support the final conclusions. This nice work opens an avenue to achieve the synergistic optimization of both the electrical and thermal transports. Therefore, I recommend this manuscript for publication in *Nature Communications* after addressing my minor concerns.

1. The authors emphasized the electrical transport properties of the rhombohedral GeTe in fig.2. It is suggested that DSC data should be collected to verify the accurate phase transition temperature.

Response: Thanks for your helpful suggestion. The DSC data are displayed in **Supplementary Fig. 4.**

Supplementary Figure 4. DSC measurement. Differential scanning calorimetric (DSC) measurements performed on (a) B0/BGT and (b) B10/BGT samples.

2. In fig.3, the thermal conductivity decreases as the boron content increases. However, the lattice thermal conductivity decreases first, followed by a rise. Could you please explain why the lattice thermal conductivity exhibits such a variation trend?

Response: Thanks for your good question. The reduced lattice thermal conductivity is mainly attributed to the presence of strain-induced dislocations (Figure 4), which play an important role in scattering the mid-frequency phonons. However, due to the high thermal conductivity of boron inclusions, the lattice thermal conductivity increases with increasing boron content.

3. Please provide the corresponding voltage-current (V-I) plot to support the output power and efficiency data of the device.

Response: Thanks for your suggestion. The corresponding voltage-current plot is displayed in Supplementary Fig. 33.

Supplementary Figure 33. V-I plot. The tested voltage-current relationship.

4. It is evident that there is a difference in the energy conversion efficiency between the simulation data and experimental data of the device. Could you please give some comments?

Response: Thanks for your good question. The calculation of theoretical efficiency through the COMSOL simulation does not take the contact resistance, heat radiation and heat convection into account¹. As a result, the calculated output power and current were different from the experimental results.

Reviewer #2 (Remarks to the Author):

The manuscript reports on the thermoelectric properties of the (Ge,Bi)Te + B samples. While there are a lot of good data, the data analysis and representation are of sufficient quality for the Nature Communication.

1. The title itself and abstract are misleading. First of all, the samples are not nanocomposites. The prepared samples are polycrystalline, with the submicrometric inclusions of B and Ge impurities (>200 nm). Secondly, GeTe is doped with Bi, which improves the properties significantly. This is not stated in the title nor mentioned in the abstract; such omission creates an illusion that B does all the work.

Response: Thanks for your helpful suggestions. The title and abstract were revised as follows by changing “nanocomposite” to “composite”. Indeed, Bi doping also has positive effect on manipulation of thermoelectric properties in our work, so we also modified the title by adding “GeTe-based thermoelectric” before “composite”. And to avoid overlapping, we deleted “thermoelectric” before “figure of merit”.

Title: Exceptional figure of merit achieved in boron-dispersed GeTe-based thermoelectric composites.

Abstract: In this work, the TE performance of GeTe is improved by a facile composite approach. We find that incorporating a small amount of boron particles into the Bi-doped GeTe leads to significant enhancement in power factor and simultaneous reduction in thermal conductivity.

Furthermore, the thermal mismatch between the boron particles and the matrix induces high-density dislocations that effectively scatter the mid-frequency phonons, accounting for a minimum lattice thermal conductivity of $0.43 \text{ Wm}^{-1}\text{K}^{-1}$ at 613 K. Consequently, we obtain a maximum figure of merit Z_{max} of $4.0 \times 10^{-3} \text{ K}^{-1}$ at 613 K in the GeTe-based composites, which is the record-high value in GeTe-based TE materials and also superior to most of TE systems for mid-temperature applications.

2. Analysis of the scattering parameter, r , in Eq. 3 on page 11 is questionable. Both the r and effective mass, m^* , will influence the Seebeck coefficient, and thus one cannot establish r without knowing m^* . The authors refer to Ref. 73, where this issue of uncertainty of r and m^* was stated. The authors appeared to ignore this and decided to derive r from S , without really stating what value of m^* was used and why.

Response: Thanks for your good question. According to the DFT calculations (Supplementary Fig. 24), the results show that Bi doping has negligible effects on band shape, indicating that the band effective mass (m_b^*) is not significantly modified. However, the energy separation between the valence band maxima at the L and Σ points decreases from 174 meV to 74 meV after Bi doping. According to the formula, $m_d^* = N_v^{2/3} \cdot m_b^*$, the improvement of DOS effective mass is reasonable.

According to the Rietveld refinement, the addition of boron inclusions has negligible effects on lattice parameters (especially the interaxial angle). Thus, the boron inclusions hardly affect the m^* . For a fixed m^* , the scattering factor can be figured out according to the Eq. 3:

where k_B is the Boltzmann constant, e is the electron charge, h is the Planck constant, T is the

temperature, n is the carrier concentration and r is the scattering factor.

The scattering of carriers of the Bi-doping sample (namely, the B0 sample) is dominated by the acoustic phonons. Thus, the r is fixed to be -0.5, the m^* can be figured out to be $1.72m_0$. In the following boron addition, the m^* is fixed, and the r can be figured out, shown in **Supplementary Table 3.**

Supplementary Figure 24. Band structure. DFT calculated band structure for (a) Ge₁₈Te₁₈ and (b) BiGe₁₇Te₁₈.

Mechanism Analysis. To better understand the change in electrical transport properties, the DFT calculations were carried out to examine the electronic band structure via Bi doping (Supplementary Fig. 24). Both the s orbital energy of the dopants and the interaxial angle are key to inducing the band convergency⁵⁵. According to the previous results²³, the s orbital energy of Bi does not obviously contribute to the band convergency. Considering the changes in crystal structure induced by Bi (Supplementary Table 2), the valance band convergency is promoted, supported by our DFT calculation results. After Bi doping, the energy separation between the valence band maxima at the L and Σ points decreases from 174 meV to 74 meV. As a result, Bi doping improves the m^* and S . As for boron addition, there are negligible effects on lattice parameters (especially the interaxial angle, Supplementary Table 2), and boron atoms hardly enters the matrix lattice. Therefore, the interface should be responsible for the phenomenon in Figure 2d.

(Eq. 3) ⁷³, is used to further figure out the carrier scattering factor, where e is the electron charge, h is the Planck constant and n is the carrier concentration. The carrier scattering in the Bi-doped sample (the B0/BGT sample) is usually dominated by the acoustic phonons. Thus, r is fixed to be -0.5, and the m^* can be determined to be $1.72m_0$ based on the SPB model. As for the boron-added samples, the r can be estimated given a fixed m^* , and the r can be figured out, shown in **Supplementary Table 3**.

3. The choice of the additional sintering temperature of 723K is not clear. Yes, this temperature will not promote Ge dislocation as the temperature is low and external pressure is large.

Despite this, the data in Fig. S 22 and 23 for the BGT-723 and BGT/B40-723 are very useful especially in comparison to the data in Fig 3b and 4 for BGT and BGT/B40. From this comparison, one can deduct that the primary contribution to the reduction in the lattice thermal conductivity is formation Ge dislocations, and not the B inclusions.

Both samples have the same amount of B inclusions and the only difference between them is the annealing temperature and the resulting Ge vacancy dislocations. Thermal mismatch between the boron particles and the GeTe matrix should be relatively similar for the two sintering temperatures.

Response: Thanks for your comments. Different sintering temperatures indeed have a significant effect on the resulting dislocations. On one hand, the higher sintering temperatures may promote the evolution of vacancies. On the other hand, the higher sintering temperatures may induce higher strains due to the higher temperature drop (Eq. 2). The relevant description is revised as follows.

Through the comparison of the samples with the same boron content sintered at different temperatures (**Figure 3** and **Supplementary Figs. 28-29**), it can be deducted that the primary contribution to the reduction in κ_L is the formation of dislocations. Because the two samples have the same amount of boron inclusions, the formation of dislocations should be mainly attributed to the sintering temperature. On one hand, the higher sintering temperatures may promote the evolution of vacancies. On the other hand, the higher sintering temperatures may induce higher strains due to the higher temperature drop (Eq. 2).

4. The entire narrative of the low thermal conductivity due to the B inclusions is not valid. Other researchers reported similar thermal conductivities in the samples that do not have B inclusions. For example, <https://doi.org/10.1002/adma.202008773>, <https://doi.org/10.1021/jacs.8b12624>.

Response: Thanks for your kind suggestion. The relevant description is revised as follows.

Furthermore, \$\kappa_L\$ is suppressed because the mid-frequency phonons are scattered by the strain-induced by high-density dislocations. According to the calculation results, the dislocations play a major role in scattering the mid-frequency phonons compared to the inclusions, which is consistent with our observations.

5. There is no convincing argument proving that the increase in thermopower stems from the “the interfacial potential barriers” and not from the increase in the carrier effective mass related to the changes in the electronic structure. The Pisarenko plot in Figure 2d points to the increase in the effective mass. In the same figure, the authors indicate that the scattering parameter r increases, but as discussed above this is questionable.

Response: Thanks for your good question. As we discussed above (in question 2), the introduction of Bi improves the band convergency, thereby increasing \$m^*\$. According to the Rietveld refinement, the addition of boron inclusions has negligible effects on lattice parameters (especially the interaxial angle). Thus, the boron inclusions hardly affect the \$m^*\$. For a fixed \$m^*\$, the scattering factor can be figured out according to the Eq. 3. Therefore, the scattering of carriers can be attributed to the interfacial potential barrier, which blocks part of holes, improving the Seebeck coefficient and thereby PF.

6. Figure 2e. Please, show data for the $\text{Ge}_{0.9}\text{Mg}_{0.04}\text{Bi}_{0.06}\text{Te}$ sample till 700K. <https://doi.org/10.1002/adma.202008773>

Response: Thanks for your suggestion. **Figure 2e** shows the comparison of power factor for samples in rhombohedral phase. The data for samples in the entire temperature range is shown in **Supplementary Fig. 3**.

Supplementary Figure 3. Seebeck coefficient and power factor measurement. Temperature dependence of (a) Seebeck coefficient and (b) power factor for BGT/B samples. (c) The comparison in power factor and (d) average power factor between B/BGT samples (B0/BGT and B10/BGT samples) with the samples in literatures^{14,15,21–23}.

7. Figure 3e. Please, include the data for $\text{Ge}_{0.86}\text{Sb}_{0.1}\text{Zn}_{0.04}\text{Te}$.
<https://doi.org/10.1021/jacs.8b12624>

Response: Thanks for your suggestion. We have added the data in Figure 3e.

59. Hong, M. *et al.* Strong Phonon–Phonon Interactions Securing Extraordinary Thermoelectric $\text{Ge}_{1-x}\text{Sb}_x\text{Te}$ with Zn-Alloying-Induced Band Alignment. *J. Am. Chem. Soc.* **141**, 1742–1748 (2019).

Figure 3 | Phonon transport properties and figure of merit ZT value. Temperature dependence of (a) total thermal conductivity and (b) lattice thermal conductivity of $\text{Bi}_{0.05}\text{Ge}_{0.94}\text{Te}-y$ wt. % B samples. (c) The comparison of the BGT/B0 and BGT/B10 samples in the lattice thermal conductivity, sound velocity and mean free path of phonons. (d) ZT values of $\text{Bi}_{0.05}\text{Ge}_{0.94}\text{Te}-y$ wt. % B samples. Comparison of (e) the maximum ZT value and (f) the average ZT value of GeTe-based materials in this work with lead-free TE materials^{20,23,40,41,56-59} and Pb-doped TE materials^{25,55,60,61}.

8. $\text{Bi}_{0.05}\text{Ge}_{0.94}\text{Te}$ and $\text{Bi}_{0.05}\text{Ge}_{0.94}\text{Te}+\text{B}$ are also cation excessive. The charge balanced composition is $\text{Bi}_{0.05}\text{Ge}_{0.925}\text{Te}$.

Response: Thanks for your valuable comment. The relevant description is revised as follows.

In order to figure out the main reason for reduced \$\kappa_L\$, the \$\text{Bi}_{0.05}\text{Ge}_{0.96}\text{Te}\$ and \$\text{Bi}_{0.05}\text{Ge}_{0.96}\text{Te}-0.1\$ wt. % B samples (less Ge-deficiency samples) were fabricated.

9. The authors should report on the amount of each sample made. This is important in the light of the B amount used.

Response: Thanks for your helpful suggestion. The relevant description is revised as follows.

Next, the powders (\$\text{Bi}_{0.05}\text{Ge}_{0.94}\text{Te}\$ ) were mixed with amorphous boron (powder, 99%, Aladdin) via MA in a planetary ball mill at 300 rpm for 2 h to fabricate a series of boron-added samples (\$\text{Bi}_{0.05}\text{Ge}_{0.94}\text{Te}-y\$ wt. % B samples, in which \$y = 0.00, 0.05, 0.10, 0.20, 0.40\$ ). The total mass of

powders for one jar is 10 g, in which the addition amount of boron powder is 0 g, 0.005 g, 0.01 g, 0.02 g and 0.04 g for Bi_{0.05}Ge_{0.94}Te- γ wt. % B samples where $\gamma = 0.00, 0.05, 0.10, 0.20, 0.40$, respectively. Then, the obtained powders (about 7 g) were densified by spark plasma sintering (SPS 211Lx, Fuji Electronic, Japan) at 873 K for 5 min under a pressure of 60 MPa.

Reviewer #3 (Remarks to the Author):

In the manuscript by Jiang et al., a high thermoelectric figure of merit (zT) for a GeTe-based composite is reported, along with impressive one-leg conversion efficiency, attributed to an extremely modest addition of boron. Despite the noteworthy results, the employed strategy of nanocompositing and the claim of Bi-GeTe achieving a peak zT exceeding 2 are not novel. Consequently, the work may not be suited for a high-profile publication like Nature Communications but could find a home in a more specialized materials science journal.

Before considering resubmission, the authors should resolve some inconsistencies:

Response: Thanks for your comments. In fact, the core concept of our work is that the difference in the thermal expansion coefficient between the boron inclusions and GeTe matrix, induces strain fluctuations near the interfaces, leading to the formation of high-density dislocations. These defects induced by boron inclusions enhance the phonon scattering, thereby significantly reducing the lattice thermal conductivity. This mechanism is revealed in detail, providing new insight into thermal transport regulation. Furthermore, an additional benefit of this strategy is that the interfacial potential barrier between inclusions and matrix can be optimized simultaneously, blocking part of holes and modulating the carrier scattering factors. Benefiting from the significantly reduced lattice thermal conductivity and the enhanced power factor, a record high Z value is eventually achieved in GeTe-based thermoelectric materials.

Supplementary Figure 1. Z value. The comparison in Z values between the BGT/B10 sample and the samples in literatures¹⁴⁻²⁰.

1. The manuscript emphasizes the role of Bi as a phonon scattering center that alters band mass. If so, detailed theoretical band structure calculations, especially changes in effective mass after Bi incorporation, should be presented. The effective mass derived from the Pisarenko plot in Figure 2d appears to be merely an empirical fit.

Response: Thanks for your suggestion. According to the DFT calculations (Supplementary Fig. 24), the results show that Bi doping has negligible effects on band shape, indicating that the band effective mass (\$m_b^*\$ ) is not significantly modified. However, the energy separation between the valence band maxima at the L and \$\Sigma\$ points decreases from 174 meV to 74 meV after Bi doping, improving band degeneracy (\$N_v\$ ). According to the formula, \$m_d^* = N_v^{2/3} \cdot m_b^*\$, the improvement of DOS effective mass is reasonable (Figure 2d).

Supplementary Figure 24. Band structure. DFT calculated band structure for (a) Ge₁₈Te₁₈ and (b) BiGe₁₇Te₁₈.

To better understand the change in electrical transport properties, the DFT calculations were carried out to examine the electronic band structure via Bi doping (Supplementary Fig. 24). Both the s orbital energy of the dopants and the interaxial angle are key to inducing the band convergency⁵⁵. According to the previous results²³, the s orbital energy of Bi does not obviously contribute to the band convergency. Considering the changes in crystal structure induced by Bi (Supplementary Table 2), the valance band convergency is promoted, supported by our DFT calculation results. After Bi doping, the energy separation between the valence band maxima at the L and Σ points decreases from 174 meV to 74 meV. As a result, Bi doping improves the m^* and S. As for boron addition, there are negligible effects on lattice parameters (especially the interaxial angle, Supplementary Table 2), and boron atoms hardly enters the matrix lattice. Therefore, the interface should be responsible for the phenomenon in Figure 2d.

2. While the carrier concentration remains constant at 300 K (Figure 2c), a notable decrease in the Seebeck coefficient for B0 and B5 at the same temperature is observed. Clarification on this discrepancy is required.

Response: Thanks for your helpful suggestion. While the carrier concentration remains constant at 300 K, the Seebeck coefficient increases with increasing boron content. This phenomenon can be attributed to the enhanced scattering factor. Because there are negligible effects on lattice

parameters (especially the interaxial angle, **Supplementary Table 2**), and boron atoms hardly enters the matrix lattice, the changes to m^* are negligible after adding boron inclusions. However, the interface performs as a barrier to block part of holes and modulate the carrier scattering factors. According to the Eq. 3, _____, the Seebeck coefficient increased after adding boron inclusions. The B0/BGT and B5/BGT samples have lower boron content, and the interface scattering is weaker, showing the lower scattering factor and Seebeck coefficient.

3. The zT curves for B-GeTe (Figure 3d) deviate from typical behavior in GeTe systems. The peak values coincide with phase transition temperatures, raising concerns about structural stability, as the highest performance is precariously near the phase transition. Moreover, there is a precipitous performance drop after the peaks are reached.

Response: Thanks for pointing out this critical issue. The relevant differential scanning calorimetric (DSC) tests were carried out to check the temperature range of the phase transition² for the B0/BGT and B10/BGT samples, as shown in **Supplementary Fig. 4**. All data are deleted in the temperature range of phase transition. The relevant description is also revised as follows.

Abstract

Furthermore, the thermal mismatch between the boron particles and the matrix induces high-density dislocations that effectively scatter the mid-frequency phonons, accounting for a minimum lattice thermal conductivity of $0.43 \text{ Wm}^{-1}\text{K}^{-1}$ at 613 K. Consequently, we obtain a maximum figure of merit Z_{max} of $4.0 \times 10^{-3} \text{ K}^{-1}$ at 613 K in the GeTe-based composites, which is the record-high value in GeTe-based TE materials and also superior to most of TE systems for mid-temperature applications.

Introduction

Furthermore, κ_L is suppressed because the mid-frequency phonons are scattered by the strain-induced high-density dislocations. Due to the synergistic optimization of carrier and phonon transport, the boron-added samples obtain an extremely high ZT value of 2.45 in R-GeTe compared to the samples prepared by the traditional doping methods (**Figure 1b**). The maximum figure of merit ($Z_{\text{max}} = 4.0 \times 10^{-3} \text{ K}^{-1}$) of synthesized GeTe-based material is the

record-high value in GeTe-based TE materials, and competitive among the TE materials for medium temperature applications, which is more intuitive to evaluate the transport properties without temperature factor (Figure 1c, Supplementary Fig. 1).

Results

Electrical Transport.

Notably, the S increases from $83.14 \mu\text{VK}^{-1}$ to $97.3 \mu\text{VK}^{-1}$ at 300 K as the boron content increases from 0 to 0.40 wt.%; the B0/BGT and B40/BGT samples reach maximum S values of $226.5 \mu\text{VK}^{-1}$ and $259.8 \mu\text{VK}^{-1}$ at 613 K, respectively (Figure 2b).

Thermal Transport and ZT value.

As a result, a high ZT value of 2.45 is achieved in the rhombohedral B10/BGT sample at 613 K (Figure 3d). Notably, our work also demonstrates good repeatability ($ZT = 2.44$ and $Z = 4.0 \times 10^{-3} \text{K}^{-1}$ in Supplementary Fig. 7), showing much higher ZT values than most of other GeTe-based TE materials (Figure 3e). In addition, boron-added samples attain a high average ZT of 1.1 in the temperature range from 300-613 K.

Consequently, a minimum κ_L of $0.43 \text{Wm}^{-1}\text{K}^{-1}$ is achieved in the B10/BGT sample, approaching the theoretical minimum κ_L of GeTe following the Clarke model.

Supplementary Figure 4. DSC measurement. Differential scanning calorimetric (DSC) measurements performed on (a) B0/BGT and (b) B10/BGT samples.

Figure 2 | The electrical transport properties.

Figure 3 | Phonon transport properties and figure of merit ZT value.

4. The claim of achieving a minimum lattice thermal conductivity (KL) of 0.37 Wm-1K-1 in the B10/BGT sample, nearing the theoretical minimum KL for GeTe, necessitates verification via measured heat capacity data, particularly because of the boron composite's nature.

Response: According to your suggestion, we tested the C_p values for the B0/BGT and B10/BGT samples, as shown in **Supplementary Fig. 8**, indicating a slightly higher C_p value of the B10/BGT sample than that of the B0/BGT sample, due to the light boron atoms. The calculated temperature-dependent ZT values are shown in **Figure R1**. The minimum lattice thermal

conductivity is 0.43 and 0.41 W/mK at 613 K calculated by the Dulong-Petit C_p and DSC measured C_p , respectively.

Supplementary Figure 8. C_p measurement. The C_p value for (a) the B0/BGT sample and (b) the B10/BGT sample.

Figure R1. The temperature-dependent ZT values for the B0/BGT sample and the B10/BGT sample calculated by the Dulong-Petit C_p and DSC measured C_p , respectively

5. When assessing conversion efficiency, the potential for thermal radiation losses during heat flow measurement with mini-pem must be considered and discussed, as this is a common measurement challenge.

Response: In this instrument, the heat flow is evaluated by the flow calorimeter. In the flow calorimeter, the temperature difference between the inlet temperature in liquid (T_{in}) and the

outlet temperature in liquid (T_{out}), and the velocity in liquid (v) depend on the heat flow in the sample. However, due to the small size of our sample (around $2.8 \times 3 \text{ mm}^2$) compared to the large dimension of Cu heating block attached to the sample ($10 \times 10 \text{ mm}^2$) and the high measurement temperature (over 300 degC), there is much heat loss from the heating block to the cooling block in the form of thermal radiation (**Figure R2a**). Considering this amount of heat loss, the amount of heat loss is calculated using the sample thickness. The calculated heat loss amount is based on the condition without the sample (**Figure R2b**), so it is calculated using the percentage of the contact surface area of the copper block that does not contact the sample. Regarding the heat radiation txt file, the company (Advanced Riko, Inc.) used data measured with the distance of 2 mm, 5 mm, and 10 mm between the heating block and cooling block, where no samples are placed. Using these results, the amount of heat loss at each set temperature can be figured out.

Figure R2. The schematic diagram of the mini-PEM device a) with the sample and b) without sample.

Due to the small size of the device (about $2.8 \text{ mm} \times 3 \text{ mm}$ for the cross-sectional area) and high testing temperature (over 573 K), the Q is revised according to the analysis system provided by the company.

6. Supplementary Figure S21e & f present Ge concentrations of 28 & 36 at% Ge, which seem implausible. It is recommended to corroborate these findings with low magnification SEM

images in backscattered electron (BEI) mode, supplemented by WDS analysis to authenticate elemental distribution.

Response: We conduct EPMA and TEM characterizations on the samples. **Figure R2** shows the back scattered electron images for the B0/BGT and B40/BGT samples. As for the B0/BGT sample, the dark contrast area is Ge-rich; the dark contrast area can be the gathering of Ge or B inclusion in the B40/BGT sample, as shown in **Supplementary Figs. 12-13**. The actual chemical composition of the boron-added sample is also shown in **Table R1**, which indicates that the average Ge concentration is around 42 at.%. We further used the STEM/EDS characterizations to check the Ge concentration, as shown in **Figure R3** and **Table R2**. The area with high-density dislocations also shows low Ge concentration. This difference can be attributed to the different electron beam size between the EPMA (0.5-1 μm) and STEM (~ 20 nm). Due to the large electron beam size in EPMA, more grains will be examined and the signals of the grains with limited Ge vacancies may be included, which could be responsible for the differences in Ge concentrations.

Supplementary Figure 13. Back scattered electron imaging characterization. The back scattered electron imaging for a) the B0/BGT sample and b) the B40/BGT sample.

Table R1. Actual chemical composition of for the B40/BGT sample tested by EPMA.

Nominal composition	Ge	Te	Bi
#1	42.0576	55.0569	2.8855
#2	41.6771	55.529	2.7939
#3	42.2283	55.0561	2.7155
#4	42.2773	55.0009	2.7218

#5	41.9571	55.2148	2.8281
Average	42.0395	55.1715	2.7890

Figure R3. The STEM images for the boron-added sample.

Table R2. The STEM/EDS data for the boron-added sample.

Nominal composition	Ge	Te	Bi
#1	40.62	57.01	2.37
#2	41.00	56.96	2.04
#3	32.97	65.06	1.97
#4	33.67	64.03	2.31
#5	30.56	66.43	3.01

7. Inclusion of high-temperature (in-situ) powder X-ray diffraction (PXRD) data for the best-performing samples is advisable, given that the optimal performance region is significantly above room temperature.

Response: Thanks for your kind suggestion. The high-temperature PXRD test was conducted, as shown in Supplementary Fig. 11. The peaks of polycrystalline Al₂O₃ are shown in the patterns. The sample holder is Al₂O₃ ceramic, and the edge of the holder can be detected by the x-ray. The Ge peaks are detected after 623 K, which is attributed to Ge precipitates generated in the high vacuum (10⁻⁴ - 10⁻⁵ mbar) during the in-situ high-temperature test. The tests of

thermoelectric properties are stable, as shown in Figure R4. The XRD patterns before testing and after testing of the sample is shown in Figure R5.

Supplementary Figure 11. Phase characterization in high temperatures.

Temperature-dependent x-ray diffraction patterns of (a) the BGT/B0 sample, (b) BGT/B10 sample. The sample holder is Al₂O₃ ceramic, and the edge of the holder can be detected by the x-ray. As a result, the peaks of polycrystalline Al₂O₃ are shown in the pattern. The Ge peaks are detected after 623 K, which is attributed to Ge precipitates generated in the high vacuum (10⁻⁴ - 10⁻⁵ mbar) during the high-temperature test.

Figure R4. The cycling tests for the B10/BGT sample.

Figure R5. The XRD patterns for the sample before and after testing.

References

1. Roychowdhury, S. *et al.* Enhanced atomic ordering leads to high thermoelectric performance in AgSbTe₂. *Science* **371**, 722–727 (2021).
2. C. Liu, Z. Zhang, Y. Peng, F. Li, L. Miao, E. Nishibori, R. Chetty, X. Bai, R. Si, J. Gao, X. Wang, Y. Zhu, N. Wang, H. Wei and T. Mori, *Sci. Adv.*, 2023, **9**, eadh0713.

Sincerely Yours,

Jing-Feng Li, Professor

REVIEWER COMMENTS

Reviewer #1 (Remarks to the Author):

It is well revised, so the manuscript could be accepted.

Reviewer #2 (Remarks to the Author):

The manuscript is suitable for the publication.

Reviewer #3 (Remarks to the Author):

While I acknowledge the significant efforts Jiang et al. have made in revising the paper, several critical issues still need to be addressed:

1. The use of GeTe doped with Bi is not a novel approach, as it is well-documented for its high figure of merit. Although the authors have introduced Boron into the mix, the material system essentially remains within the category of nanocomposite GeTe, which does not significantly enhance its novelty.
2. The authors present a high zT value and propose a mechanism supported by some experimental evidence. However, the work primarily aims to highlight an exceptionally high conversion efficiency, which necessitates further emphasis.
3. Accurate conversion efficiency data is crucial. The manuscript lacks heat flow data, which is essential for validating the efficiency claims. Additionally, the V-I slope inconsistency in Figure S33 (at a ΔT of 421 K) raises concerns about the reliability of the data.
4. The use of the commercial instrument (minipem) to collect output power and heat flow data is commendable as it provides a standard comparison framework. However, it is imperative in the revision that the authors include screenshot images of the best-performing output power (P) and heat flow (Q) from the minipem software to substantiate their claims.

Dear Editor and reviewers,

Thank you very much for your decision letter and review reports on our manuscript entitled “**Exceptional figure of merit achieved in boron-dispersed GeTe-based thermoelectric composites**” (Manuscript ID: NCOMMS-24-11820). We greatly appreciate the valuable comments from reviewers. We hope the revised manuscript will address the concerns raised by the third reviewer and meet the requirements of your esteemed journal *Nature Communications*. Our point-to-point responses are listed as below:

Answers to reviewers:

Reviewer #1 (Remarks to the Author):

It is well revised, so the manuscript could be accepted.

Response: Thanks for your recognition of our revised paper, and your positive attitude towards publication.

Reviewer #2 (Remarks to the Author):

The manuscript is suitable for publication.

Response: Thanks for your recognition of our revised paper, and your positive attitude towards publication.

Reviewer #3 (Remarks to the Author):

While I acknowledge the significant efforts Jiang et al. have made in revising the paper, several critical issues still need to be addressed:

1. The use of GeTe doped with Bi is not a novel approach, as it is well-documented for its high figure of merit. Although the authors have introduced Boron into the mix, the material system essentially remains within the category of nanocomposite GeTe, which does not significantly enhance its novelty.

Response: Thank you! We understood your concerns about the novelty of Bi doping for GeTe, but which is not the core concept of this work.

Yes, the material system remains within the category of nanocomposite GeTe, but we like to emphasize the following points:

1) Quite few investigations on GeTe composites were conducted in the last decade, although the effects of different dopants on TE performance of GeTe-based thermoelectric materials were extensive (*Adv. Mater.* **2023**, **35**, **2208272**, *Adv. Funct. Mater.* **2024**, **2403498**). Since the ZT values achieved by doping are already very high, it is difficult to find an effective compositing approach to further increase it, but we did realize it by mixing boron particles in this work, leading to a higher Z than reported for GeTe.

2) Even for thermoelectric composites, we would like to draw your attention to the underlying mechanisms of electrical and thermal transport modulation, which is different from the ordinary nanocomposite strategy focusing on enhancing phonon scattering via traditional nanoparticle scattering. We revealed that the boron inclusions are able to induce strain fluctuations near the interfaces and promote the dislocation evolution, mainly due to the mismatched thermal expansion coefficient between the inclusions and matrix; this enhanced phonon scattering essentially originates from the high-density dislocations. Importantly, as an additional benefit, the boron inclusions can modulate the interfacial potential barriers and hence the carrier scattering factor, which helps to improve the Seebeck coefficient and PF.

2. The authors present a high zT value and propose a mechanism supported by some experimental evidence. However, the work primarily aims to highlight an exceptionally high conversion efficiency, which necessitates further emphasis.

Response: Thanks for your comments. Here, we would like to clarify that our work aims to highlight an exceptionally high Z value achieved in the GeTe-based material rather than the high thermoelectric conversion efficiency of the as-prepared device. As you know, the efficiency can be affected by many factors. For example, one of them is the interface electrical and thermal resistances between the thermoelectric leg and electrode. That means the efficiency of device is not solely related to the Z or ZT value. Notably, the Z value is exactly a key parameter for evaluating the thermoelectric properties of materials, because the high ZT values may originate from the temperature factor and thus the electron and phonon transports can be intuitively evaluated at different temperatures using Z instead of ZT (*Science* 375, 1385–1389 (2022)). Actually, we prepared a single-leg device in order to show the application potential of our optimized material, whose values are quite high as shown in Figure 5. According to your suggestions, we also revised the related part in the text.

3. Accurate conversion efficiency data is crucial. The manuscript lacks heat flow data, which is essential for validating the efficiency claims. Additionally, the V-I slope inconsistency in Figure S33 (at a ΔT of 421 K) raises concerns about the reliability of the data.

Response: Thanks for your valuable comments. We agree that the heat flow data are important. According to your helpful suggestions, we carefully checked the heat flow data of the best-performing sample, and found that the data points showed some fluctuations, which may bring about some deviations (Figure R1); this unstable heat flow could be related to the purity of the circulating water. In order to minimize the uncertainties and validate the high conversion efficiency, we prepared a new single-leg device and carried out the measurement again. At this time, the heat flow data seem more normal and the corresponding conversion efficiency is 13.7%, close to 14%, as reported in our original version.

The increasing internal resistance with increasing temperature difference, as shown in Figure R2, is common in device/module tests due to the additional interfacial thermal resistance (*Energy Environ. Sci.*, 2021, 14, 6506–6513, *Energy Environ. Sci.*, 2023, 16, 6147–6154).

Figure R1. The current-dependent heat flow for the two-segmented device.

Figure R2. The internal resistance R for the device.

4. The use of the commercial instrument (minipem) to collect output power and heat flow data is commendable as it provides a standard comparison framework. However, it is imperative in the revision that the authors include screenshot images of the best-performing output power (P) and heat flow (Q) from the minipem software to substantiate their claims.

Response: Thanks for your concerns. Based on your valuable comments, we provide the screenshot images of the best-performing output power (P) and heat flow (Q) from the minipem software for the newly fabricated single-leg device, as mentioned in response to Q3. Please check them.

As a result, our newly fabricated single-leg device can still achieve a high conversion efficiency of 13.7%; the slight difference in conversion efficiency should be also within the margin of uncertainties. In order to meet the high standard of data quality in **Nature Communications**, we update the relevant data based on the new measurement results.

Figure 5 | The stimulation and measurement results of the single-leg thermoelectric device.

(a) Contour map of efficiency (η) of GeTe/(Bi,Sb)₂Te₃ segmented TE leg when $T_h = 723$ K and $T_c = 300$ K. **(b, c)** The tested conversion efficiency and output power, respectively. The inset image in (c) showing the schematic diagram of the segmented TE leg. **(d)** The comparison of

the conversion efficiency with the results in literatures^{17,70,77,78}.

Supplementary Figure 33. Mini-PEM measurement. The tested voltage-current relationship and heat flow.

As shown in Figure 5b, the highest conversion efficiency (η_{\max}) of 13.7% is yielded ($\Delta T = 455.9$ K). When $\Delta T = 455.9$ K and $I = 3.79$ A, the maximum output power exceeds 0.18 W (Figure 5c), and the corresponding V-I relationship and heat flow are shown in Supplementary Fig. 33.

Figure R3. The V-I plot, output power (P) and heat flow (Q) for the testing sample.

Sincerely Yours,
 Jing-Feng Li, Professor
 On behalf of all authors

REVIEWERS' COMMENTS

Reviewer #3 (Remarks to the Author):

The manuscript is well-revised, and it is recommended to publish.

Answers to reviewers:

Reviewer #3 (Remarks to the Author):

The manuscript is well-revised, and it is recommended to publish.

Response: Thanks for your recognition of our revised paper, and your positive attitude towards publication.

Sincerely Yours,

Jing-Feng Li, Professor
